# Timing and Graded BMP Signalling Determines Fate of Neural Crest and Ectodermal Placode Derivatives from Pluripotent Stem Cells

**DOI:** 10.3390/biomedicines12102262

**Published:** 2024-10-04

**Authors:** Keshi Chung, Malvina Millet, Ludivine Rouillon, Azel Zine

**Affiliations:** 1LBN, Laboratory of Bioengineering and Nanoscience, University of Montpellier, 34193 Montpellier, France; 2Harvard Medical School, Massachusetts Eye and Ear Infirmary, Boston, MA 02114, USA

**Keywords:** bone morphogenetic proteins, human pluripotent stem cells, human cell models, organoids, pre-placodal ectoderm, otic lineage

## Abstract

Pluripotent stem cells (PSCs) offer many potential research and clinical benefits due to their ability to differentiate into nearly every cell type in the body. They are often used as model systems to study early stages of ontogenesis to better understand key developmental pathways, as well as for drug screening. However, in order to fully realise the potential of PSCs and their translational applications, a deeper understanding of developmental pathways, especially in humans, is required. Several signalling molecules play important roles during development and are required for proper differentiation of PSCs. The concentration and timing of signal activation are important, with perturbations resulting in improper development and/or pathology. Bone morphogenetic proteins (BMPs) are one such key group of signalling molecules involved in the specification and differentiation of various cell types and tissues in the human body, including those related to tooth and otic development. In this review, we describe the role of BMP signalling and its regulation, the consequences of BMP dysregulation in disease and differentiation, and how PSCs can be used to investigate the effects of BMP modulation during development, mainly focusing on otic development. Finally, we emphasise the unique role of BMP4 in otic specification and how refined understanding of controlling its regulation could lead to the generation of more robust and reproducible human PSC-derived otic organoids for research and translational applications.

## 1. Introduction

Many features of early human development can be recapitulated in vitro using pluripotent stem cells (PSCs). The embryonic specification of various domains arising from the different germ layers is achieved by the activation of complex and pleiotropic signalling pathways and inhibitors which interact with one another at critical times during development. Similarly, the addition of numerous small molecules that can either activate or inhibit signalling pathways in cultures of PSCs can lead to their specification and subsequent differentiation into cell types from any of the three germ layers: endoderm, mesoderm, or ectoderm.

Several in vitro protocols have been developed and refined in recent years for the generation of diverse cell types from human PSCs into complex tissue-like structures and organoids, including brain [1,2], cardiac [3,4,5,6], blood vessel [7], retina [8,9,10,11,12], lens [13,14,15,16], inner ear [17,18,19,20,21,22,23,24], etc. However, variations either across labs or between cell lines exist, suggesting the need for further optimisation of differentiation protocols in order to better understand the complex dynamics of signalling molecules involved in specific developmental pathways.

Bone morphogenetic proteins (BMPs) are one such class of signalling molecules that act in a timed manner across concentration gradients during development. Most BMPs are members of the transforming growth factor β (TGFβ) superfamily of ligands that play critical roles in a multitude of processes during the specification and development of nearly all tissue and cell types. Originally named for their ability to induce bone and cartilage formation [25], they have since been found to be involved in many aspects of development, such as extraembryonic and mesodermal specification [26,27], dorsoventral axis formation (reviewed in [28]), ectodermal patterning, and subsequent specification of ectodermal fates including neuronal, epidermal, and pre-placodal lineages [26,29,30,31,32]. Several types of BMP ligands (BMP1, BMP2, BMP4, BMP6, BMP7) have additional roles during early development and interact with multiple receptors (BMPR1A, BMPR1B, BMPR2) and mediators (SMAD proteins) for further specification of various cell types.

For BMP signalling in particular, endogenous levels of expression and activity within cell lines have previously been shown to affect the concentration of BMPs that are required to be added to cultures of PSCs in order to direct differentiation into specific lineages, such as otic lineages [17,18,20,24]. Different levels of BMP and Activin/Nodal signalling are also required for cardiac differentiation of many mouse and human PSC lines [33,34]. A similar observation has been made in human retinal organoids derived from induced PSCs, whereby activation of BMP4 had different effects on different PSC lines, resulting in the generation of different retinal cell types in organoids from each cell line [10]. Differing levels of endogenous BMP4 and BMP4 signalling activity have also been shown to affect differentiation of PSC lines into corneal epithelial-like cells [12]. This suggests that the effects of BMP4 are dependent on the PSC line used, and that in vitro differentiation protocols that are both robust and efficient require optimisation for each cell line.

In addition, the interplay between BMP signalling and other signalling pathways is complex and likely critical in determining cell fate. Indeed, induction of ectodermal placodal fate by exogenous BMP4 in human ES cell lines can be abolished by the addition of WNT3a and rescued when the concentration of BMP4 is increased relative to that of WNT3a [26]. Similarly, Camacho-Aguilar et al. [27] demonstrated the requirement of upregulation of WNT signalling in addition to BMP for conversion of human PSCs from the pluripotent state to mesodermal and extraembryonic fates. Importantly, they observed that the timing of BMP exposure was critical for the specification of different fates, with long and medium culture period duration of exposure to BMP4 driving extraembryonic and mesodermal fates, respectively, due to activation of endogenous WNT, while short pulses of BMP4 caused cells to remain in the pluripotent state. Indeed, recent studies have indicated that it is not necessarily the concentration of BMPs that is important for determining cell fate per se, but rather the integrated signalling level (i.e., concentration and duration) that determines cell fate. Recently, elegant experiments performed by Teague et al. [35] demonstrated that lower levels of BMP signalling over long durations resulted in differentiation similar to that of hPSCs exposed to higher signalling levels over shorter durations, highlighting that the timing of BMP signalling also needs to be taken into account when designing hPSC in vitro differentiation protocols.

## 2. Role and Function of BMP Signalling during Development

### 2.1. Dorsoventral Patterning and Ectodermal Derivatives

A gradient of BMP signalling is required for the mechanisms of dorsoventral axis determination during gastrulation (reviewed in [28]). In zebrafish, overexpression of BMP can rescue dorsalised mutants [36], while inhibiting BMP by overexpression of either human *TAPT1* or zebrafish *tapt1a/tapt1b* results in dorsalised embryos [37]. Similarly, in *Xenopus*, inhibition of BMP signalling by injection of the BMP antagonist USAG1 into embryos causes them to become more dorsalised [38], supporting a conserved role of BMPs in dorsoventral axis patterning during embryonic development. R-spondin 2 (RSPO2) has also been shown to regulate dorsoventral axis formation in *Xenopus* by antagonising BMP signalling [39,40]; whether R-spondins are similarly involved in mammalian dorsoventral patterning is not known. Later in development, this BMP gradient appears to be inverted in the ectodermal layer: epidermal specification occurs in the most dorsal region, where BMP concentration is highest; non-neural ectoderm (NNE), which gives rise to pre-placodal ectoderm (PPE), and neural crest in the ventral underlying regions where BMP concentration is lower; and neuronal cell fate where BMP concentration is the lowest [41,42,43] (Figure 1).

BMPs are required for the expression of NNE genes and PPE competence factors, and previous work with human stem cell lines has demonstrated that this expression requires transient BMP signalling at an optimal concentration for the generation of the desired cell types [26,30,44]. Blocking BMP signalling by the addition of Noggin abolished expression of PPE competence genes and induced expression of the neural precursor marker *HES5* [30], supporting the notion that a reduction of BMPs induces neuronal fate. Conversely, removal of Noggin in human PSCs undergoing neural induction using dual-SMAD inhibition resulted in induction of placodal fate at the expense of neuronal fate [14]. Neuroectodermal cells express several transcription factors that regulate their competency to respond to neural inducing signals and inhibit the effects of BMP and WNT signalling (reviewed in [45,46]). In contrast, high BMP levels have been found to induce epithelial differentiation of human ES cells, and addition of Noggin to these cells can increase the population of Nestin-positive neuroectodermal cells in culture at the expense of keratinocyte differentiation [47]. BMPs appear to block neural differentiation, possibly through induction of *DeltaNp63*, a transcriptional target of BMP signalling that can block neuronal development in zebrafish upon its forced expression in this model organism [48].

The PPE in turn gives rise to the sensory placodes of the head region, including the lens, inner ear, olfactory epithelium, etc. (reviewed in [49]). The development of the placodal structures and their subsequent tissues involves BMP signalling and will be discussed in detail below.

### 2.2. Placodal Lineages

Sensory placodes derived from the PPE reutilize many of the same signalling molecules and pathways that operate during earlier developmental stages to give rise to a diverse range of cell and tissue types, including the anterior pituitary gland, lens, olfactory epithelium, trigeminal ganglia, otic epithelium, and epibranchial neurons (reviewed in [49]). Studies in *Xenopus* and zebrafish have shown that, once the PPE has been specified, BMPs must then be inhibited by dorsally expressed BMP antagonists in order for placodal development to occur [29,50]. The expression of non-neural genes such as *DLX* and *GATA* inhibits the expression of neural genes such as *SOX2*, and vice versa, resulting in the establishment of distinct non-neural and neural boundaries [51]. PPE cells generated from human iPSCs and ES cells can be further differentiated in vitro to produce various placodally-derived cells, including trigeminal ganglia, lens fibres, and anterior pituitary hormone-producing cells [14,30].

#### 2.2.1. Lens Development

The specification of the PPE into PAX6-expressing anterior placode is required for the development of the eye. BMP induces expression of MAF, a downstream target of PAX6 that is required for the elongation of lens fibre cells and the expression of crystalline [52,53]. Experiments with human ES cell lines have shown that BMP inhibition is required for the induction of the anterior placode from PPE, and that addition of BMP4 is subsequently required for the induction of lens placode from these cells [30]. The addition of recombinant BMP4 or the inhibition of FGF signalling were both also able to induce expression of the lens precursor marker *PITX3* in human PSC-derived pre-placodal cells, which could be further differentiated into crystalline-positive cells containing mature lens fibres [14]. Lentoid bodies can also be generated from hiPSCs and ES cells through continuous stimulation with BMP4 and BMP7, followed by WNT activation [13,15,16], and have recently been employed in drug screening for cataract treatments [16].

#### 2.2.2. Olfactory Epithelium Development

The anterior placode appears to default to lens placode in the absence of additional signals such as FGF, which is required for the development of olfactory epithelium [54]. Loss of BMP signalling is also sufficient to cause prospective lens placodal cells to switch to an olfactory placodal fate [52]. Although BMPs play important roles in the development of the embryonic olfactory epithelium and bulb, where they are expressed along with their receptors [55,56,57], studies in mouse and chick embryos demonstrate that the expression of SOX2 is required to downregulate BMP4 in the developing olfactory epithelium for subsequent formation of the olfactory pit [56]. Downregulation of BMPs also appears to be required for the development of odorant-responsive olfactory sensory neurons derived from hiPSCs [58]. Nevertheless, BMPs continue to be expressed in the olfactory epithelium throughout adulthood, where they are thought to be important for adult neurogenesis in the olfactory system [57].

#### 2.2.3. Inner Ear Development

The posterior placodal region gives rise to the otic-epibranchial progenitor domain (OEPD), from which both the otic and epibranchial placodes are generated. The otic placode invaginates into the underlying mesenchyme to form the otic vesicle. This involves inhibition of BMP signalling, which is recapitulated in human pluripotent stem cell-derived otic organoids using the BMP inhibitor LDN193189 [17,18,22,23,24]. Such inhibition of BMP signalling could be mediated by LMO4, which was recently found to negatively regulate BMP2 and BMP4 signalling in the zebrafish inner ear [59].

BMP signalling is also important at later stages of otic development. In chick otic vesicles dissected from E3.5-4 embryos, the addition of recombinant BMP4 reduced the number of hair cells due to decreased proliferation of otic progenitor cells and increased cell death, while the addition of the BMP inhibitor Noggin increased the number of sensory hair cells [60]. Similarly, treatment of chick organotypic cultures with BMP4 during hair cell destruction prevented regeneration of hair cells from supporting cells, while Noggin was able to increase the number of regenerated hair cells [61]. In contrast, another study using chick otocyst cultures reported that blocking BMP signalling reduced generation of hair cells and supporting cells, and that exogenous BMP4 treatment increased the number of hair cells by downregulation of PAX2 in proliferating sensory epithelial progenitor cells [62]. It has been proposed that differences in the concentrations of BMP4 might be responsible for these discrepancies between studies, as the concentration of BMP4 is also found to affect patterning of sensory and nonsensory tissue in the mouse cochlea, with intermediate levels of BMP signalling required to increase the number of sensory hair cells [63]. Similar experiments have not yet been performed in stem cell-derived otic organoids to investigate whether modulation of BMP signalling could alter the number of hair cells (or indeed other otic cell types) produced within these 3D-cell structures.

#### 2.2.4. Epibranchial Placodes

The epibranchial placodes, derived from the posterior placode, give rise to sensory neurons in ganglia associated with the facial, glossopharyngeal, and vagal nerves. While the OEPD is routinely generated during production of otic organoids, and the generation of epibranchial-like neurons has been reported in these cultures [18,22], there are currently no known established models for specific and directed differentiation of epibranchial neurons from human pluripotent stem cells. Interestingly, development of epibranchial-like neurons (and other off-target neurons including neural crest) appears to occur earlier than otic neurons in these cell culture systems [22]. Treatment of stem cell aggregates with FGF, the TGFβ inhibitor SB431542, and the pan-BMP inhibitor LDN193189 was found to be sufficient for the generation of cells expressing posterior placodal markers including PAX8, SOX2, TFAP2A, ECAD, and NCAD, but not the otic marker PAX2 [18], suggesting it may be possible to generate epibranchial neurons separately from otic cells. Moreover, these cells could mature into BRN3A/POU4F1 and TUJ1-positive sensory-like neurons with a morphology more similar to epibranchial neurons than inner ear ganglia neurons. More directed differentiation and maturation of these neurons have not been investigated, although BMP signalling could be involved. Recent experiments in mice have found that blocking BMP signalling using LDN193189 strongly reduced the numbers of neuroblasts in epibranchial placode 1 and moderately in epibranchial placode 3 [64], suggesting a differential requirement for BMP signalling in neurogenesis in the epibranchial placodes.

#### 2.2.5. Trigeminal Neurons

The trigeminal ganglia are derived from the intermediate placode and contain neurons responsible for transmitting sensory information such as pain and temperature from the face. BMP signalling is implicated in the development of trigeminal ganglion neurons, possibly via interaction with MEGF8 [65]. Trigeminal sensory neurons have been generated from hiPSCs by initial activation of BMP signalling. In one protocol, trigeminal fate was subsequently induced by maintaining cells in N2 medium supplemented with ascorbic acid and BDNF [14], while another protocol used CHIR to activate WNT signalling followed by maturation in neurobasal medium supplemented with NGF, BDNF, and GDNF [66]. Engraftment of hiPSC-derived trigeminal ganglia into chicks and mice have shown their survival and ability to establish axonal projections to their target regions [14].

### 2.3. Tooth Development

Teeth are another ectodermally derived tissue, and their development requires reciprocal interactions between the epithelium and mesenchyme [67]. BMPs, in particular, play a role and have been shown to interact with other signalling pathways such as SHH [68] and WNT [69] during tooth development. Experiments in mice at embryonic days E14 and E15 have confirmed the expression of BMP2 in the oral epithelium, and of BMP4, BMP6, and BMP7 in both the epithelium and mesenchyme [70,71]. Uterine sensitization associated gene-1 (USAG1) is an antagonist of BMP signalling which is also expressed in the epithelium and mesenchyme during tooth formation [70,72]. Mice lacking USAG1 have an increased number of teeth (supernumerary teeth) which is due to enhanced BMP signalling [71,72], suggesting that BMPs are involved in regulating tooth number. Indeed, topical administration of BMP7 can result in partial supernumerary incisor formation in mouse dental explant cultures [70]. Modulation of BMP signalling has also been used to recover tooth development in mice [73]. By using antibodies to block USAG1 in a mouse model of tooth agenesis, Murashima-Suginami and colleagues were able to induce tooth formation in these mice.

Human ES cells have been used to generate oral ectoderm and dental epithelium following a differentiation protocol with increasing concentration of BMP4 [74]. These cells could be mixed with cultures of mouse dental mesenchyme and, when transplanted into murine hosts, were capable of forming tooth-like structures in vivo. Recently developed in vitro protocols have enabled the rapid generation of dental epithelial cells from hiPSCs in just over one week, by simultaneously inhibiting BMP signalling and activating SHH signalling to generate oral ectoderm from NNE, followed by activation of BMP and SHH pathways and inhibition of WNT signalling [75]. It is not clear why the induction of Pitx1-expressing oral epithelium required a low concentration of BMPs in one protocol and BMP inhibition in the other, although differences in endogenous BMP signalling and activity between the cell lines used in these studies may account for this discrepancy.

### 2.4. Neural Crest

BMP signalling, in conjunction with WNT and FGF, is also important for development of the neural crest [76,77,78,79], and WNT signalling appears to be key in determining whether ectodermal cells become NNE/PPE or neural crest. Neural crest cells, localised at the dorsolateral position of the neural tube, give rise to the neurons and glia of the peripheral nervous system, the enteric nervous system, as well as non-neural derivatives. Low concentrations of BMP4 in combination with WNT activation have been shown to generate SOX10-expressing neural crest cells from human PSC cultures [44]. Similarly, treatment of neural crest stem cell-like cells isolated from human skin with BMP2 and an activator of WNT signalling improves their multipotency and differentiation potential to neural crest lineage cells [80]. Conversely, in cultures of human ES cells, BMP signalling in combination with the inhibition of WNT signalling resulted in increased expression of SIX1-positive PPE cells and a reduced number of cells expressing PAX3 and SOX9 neural crest markers [26]. The low levels of BMP required for neural crest induction may be mediated by Gremlin 1, which acts as a BMP antagonist during early neural crest development, and also interacts with heparan sulfate proteoglycans during later stages of neural crest development [81].

### 2.5. Cardiac Development

BMPs act with other signalling pathways, including WNT, Nodal, and FGF, to induce early mesoderm (reviewed in [82]). Specification of later mesodermal fates, such as cardiac, requires additional BMP signalling. BMP2 and BMP4 are involved in cardiomyogenesis, with exogeneous application of either BMP2 or BMP4 proving sufficient to induce ectopic cardiomyocyte differentiation in chick embryos [83]. Experiments performed in precardiac spheroids generated from PSCs found that the specification of two separate populations of cardiac progenitor cells (termed first and second heart fields) requires BMP signalling, but that cells of the first heart field are specified via the BMP/SMAD pathway, while cells of the second heart field are specified through the SMAD-independent BMP/WNT pathway [3]. Moreover, blocking BMP signalling abolished the specification of both populations of cardiac progenitor cells, highlighting the importance of BMPs for early cardiac development. Nevertheless, modulation of WNT signalling is sufficient to generate heart organoids from PSCs, although the addition of BMP4 and Activin A was found to improve the size and vascularisation of organoids [5].

Certain cardiac structures, such as the cardiac outflow tract and aortic arch, are derived from neural crest cells (reviewed in [84]). Cardiac neural crest cells have also been proposed to contribute to regeneration of the myocardium following injury in zebrafish and mice [85,86,87]. In mice, cKit-positive cardiac neural crest cells possess full cardiomyogenic capacity and give rise to several cardiac cell types, which is a process dependent on BMP antagonism [88]. The suppression of BMP activity is also involved in fate specification of cardiac neural crest cells, via Adam19-mediated cleavage of ACVR1 and suppression of the BMP-SOX9 cascade [89]. In contrast, BMP activity is required for delamination of neural crest cells from the dorsal neural tube [90,91], via cleavage of N-cadherin allowing these cells to migrate [92]. Stem cell therapies based on cardiac neural crest cells derived from hiPSCs could offer a promising therapy for heart repair following disease or injury, but further investigation is required to better understand the processes involved in the specification of cardiac neural crest cells as distinct from other types of neural crest cells, and to determine how to differentiate these cells into the various cardiac cell types.

### 2.6. Bone

The role of BMPs in bone development, homeostasis, and remodelling has been extensively reviewed elsewhere [93,94]. Exposure of mesenchymal stem cells (MSCs) to BMP2 is able to induce osteogenic differentiation of these cells both in vitro and in vivo and promote bone formation [95,96,97,98]. Hydrogels containing BMP2 mimetics were found to induce bone formation when injected into rats, which was enhanced when these hydrogels were injected in combination with MSCs [95,97]. In addition to BMP2, BMP9 may also be important for bone formation and regeneration. Overexpression of BMP9 in MSCs increased their osteogenic potential and resulted in increased bone formation and bone mineral density when injected into rats with calvarial bone defects [99]. This BMP9-induced differentiation of MSCs towards osteogenic fate seems to require Notch signalling, as the inhibition of Notch prevents BMP9-induced osteogenic differentiation [100]. A recent study revealed that conditioned media from MSCs overexpressing BMP9 also enhanced bone repair of mouse calvarial defects, compared with media from MSCs that did not overexpress BMP9 [101], suggesting the presence of additional trophic factors released by these cells. BMP9 was additionally able to induce osteogenic differentiation in spheroids derived from human gingival stem cells [102], indicating that osteogenesis can be induced in several types of stem cells.

More recently, attempts have been made to induce bone formation from iPSCs, due to their greater proliferative and differentiation capabilities over MSCs. Bone formation has successfully been induced in hiPSCs using retinoic acid, which results in activation of BMP and WNT signalling pathways and differentiation of hiPSCs into osteoblast-like and osteocyte-like cells [103]. These cells were able to form bone tissue when injected into mice with calvarial defects, and also recapitulated the phenotype of osteogenesis-imperfecta when cultured from patient-derived iPSCs. Undifferentiated muscle-derived hiPSCs loaded onto an osteoconductive scaffold and implanted into mice can induce ectopic bone formation [104]. Analysis of the scaffolds at 15 and 30 days post-implantation revealed the absence of mRNA of human origin, suggesting that the implanted cells were able to induce bone formation via a paracrine communication. Indeed, conditioned media from these cells was able to induce expression of osteogenesis-related genes, upregulation of BMP2, BMP4, and BMP6, increased phosphorylation of SMAD 1/5/8, and the appearance of calcium-containing deposits in the extracellular matrix of cultured human MSCs. Further analysis of these undifferentiated hiPSCs in culture revealed higher expression of BMPs relative to expression in fibroblasts, with BMP2 levels being particularly elevated. It is unclear whether this high expression of BMPs is due to the muscle-derived origin of these cells, or whether hiPSCs derived from other cell types would have similarly high BMP expression. It also cannot be ruled out that the high BMP levels are a feature of the cell line that was used in the study. Further experiments in additional hiPSC lines derived from cells of different origins would help to clarify this issue.

Unlike bone formation in the rest of the body, the bone and cartilage of craniofacial structures are derived from cranial neural crest cells, a process which relies heavily on BMP signalling (reviewed in [105]). Treatment of human PSCs with BMP4 from day 8 after neural crest specification induces the expression of cranial neural crest markers such as *TFAP2A*, *MSX1*, and *DLX1* [106]. Increased BMP signalling in cranial neural crest cells has been shown to cause premature fusion of cranial sutures and skull bass deformities in mice [107,108,109]. As a result of this difference in embryonic origin, the MSCs found in cranial structures have different characteristics from those in the long bones. For instance, orofacial MSCs and iliac crest MSCs from the same donor have been found to behave differently when cultured in vitro. Orofacial MSCs proliferated more rapidly and had delayed senescence compared with iliac crest MSCs. Moreover, iliac crest MSCs were more responsive to osteogenic and adipogenic inductions than orofacial MSCs [110]. Recently, ectodermal MSCs, derived from human ES cells via a neural crest intermediate, have been compared with adult bone marrow-derived MSCs. They were found to have comparable osteogenic and chondrogenic abilities in culture, although ectodermal MSCs had greater proliferation and formed more dense osseous constructs in a rat calvarial defect model [111].

## 3. Regulation of BMP Signalling during Development

### 3.1. BMP Signalling Pathways and Downstream Effects on Gene Expression

BMPs act on their receptors, which are typically heterotetrameric complexes composed of type I and type II serine/threonine kinase receptors. Upon ligand binding, type II receptors phosphorylate type I receptors, which then activate SMAD1, SMAD5, and SMAD8 (Figure 2). These receptor-regulated SMADs pair with SMAD4 and translocate to the nucleus to influence the transcription of target genes. This signalling affects gene expression linked to cell growth, differentiation, and apoptosis, which is crucial during embryonic development [112,113]. Certain subclasses of BMPs, such as BMP4, have specific effects on developmental pathways, including those of inner ear hair cells and spiral ganglion neurons, highlighting their importance for neurosensory differentiation [62,114].

### 3.2. Endogenous Activators and Inhibitors of BMP Signalling

BMP signalling is finely tuned by endogenous molecules and its role in the differentiation of many cell types, including neural differentiation, is complex. While BMPs are generally antagonistic to neural differentiation at early stages of development, they promote the formation of autonomic and sensory neurons from neural crest progenitors at later stages. Extracellular antagonists like Noggin, Chordin, Gremlin, and Follistatin bind BMP proteins (Figure 2), inhibiting receptor interaction and modulating processes such as neural and limb development. Conversely, modulators like Twisted Gastrulation (TWSG1) can either enhance or inhibit BMP signalling depending on the developmental context. For instance, TWSG1 can enhance BMP signalling in the context of early neural development, promoting neural crest cell formation, while it can inhibit BMP signalling during limb formation to prevent excessive growth. USAG1 directly binds to BMPs to antagonise BMP signalling and has been shown to be important for tooth and kidney development [38,71,72,73]. Transmembrane anterior posterior transformation 1 (TAPT1), involved in axial skeletal patterning, causes proteasomal degradation of SMAD1/5, thereby inhibiting BMP signalling [37]. R-spondin 2 and 3 (RSPO2 and RSPO3) act as BMP antagonists by binding to the BMP receptor BMPR1A, resulting in their internalisation and degradation [39,40]. Intracellular inhibitors such as SMAD6 and SMAD7 prevent R-SMAD phosphorylation or promote receptor degradation, ensuring balanced BMP4 activity for normal development [115,116]. BMP4 specifically promotes glial differentiation while inhibiting oligodendrocyte formation, but this can be overridden by Notch signalling, which favours Schwann cell differentiation [117,118,119].

## 4. Consequences of BMP Dysregulation

Because of their diverse roles in development and differentiation of many cell types, dysregulation of BMPs, their receptors, and their endogenous modulators can have a spectrum of effects on nearly every tissue type involving all three germ layers. Indeed, BMPs are essential for development, with embryonic lethality reported in mice lacking expression of either BMP2 or BMP4 [120,121], while mice deficient in BMP7 experience eye, kidney, and skeletal patterning defects and die shortly after birth [122,123]. Expression of BMP4 is found to be strong in mouse caudal tissues, and loss of BMP4 in this region resulted in hindlimb fusion and lethality [124].

The effects of dysregulation of BMP signalling and its links to various diseases have been extensively reviewed elsewhere [125,126,127,128,129], highlighting the need for improved understanding of the roles of BMPs, their receptors, and their modulators in development and disease. Considering the importance of BMP signalling for development and differentiation of tissues, studying the effects of BMPs in whole model organisms is challenging, due to the lack of viability and early arrest of growth and development of embryos following perturbation of BMP signalling. Moreover, many different tissues and organs may be affected, which further complicates interpretation of the effects of loss, mutation, or forced expression of BMP and/or its receptors and modulators in the whole organism. Some of these effects are likely to be secondary, arising from gross defects caused by dysregulation of BMP signalling, rather than as a direct consequence of BMP signalling itself. Conditional knockouts (and other similar targeting of specific tissues) might overcome some of these limitations. For instance, while loss of BMP4 results in embryonic lethality, Suzuki et al. [124] were able to use a conditional knockout *Isl1*-Cre mouse line in which BMP4 expression was reduced in the caudal body region only, allowing their mice to survive to a developmental stage late enough to investigate the caudalising effects of BMP4. Likewise, Chang et al. [130] conditionally deleted BMP4 expression in the mouse inner ear and were able to demonstrate the importance of BMP4 for the formation of the vestibular cristae and canals. They also succeeded in electroporating expression vectors to inhibit BMP signalling directly into the otocyst of the developing chick, and observed that downregulation of BMPs resulted in patterning defects in the crista, although they cautioned that some of the effects could also be due to electroporation rather than reduced BMP signalling.

Using pluripotent stem cells to investigate BMP signalling could also be used to overcome some of the limitations mentioned above, although care must be taken to ensure that perturbations of BMP signalling do not affect their overall survival, maintenance, and differentiation potential. Indeed, the ability of cells to differentiate towards the desired lineage is likely to be affected in conditions of abnormal BMP signalling or if cells are unable to respond to exogenous BMP. The maintenance of murine ES cell pluripotency has been shown to require BMP4, which induces the expression of *Klf2* [131], although direct BMP signalling is unlikely to be involved in the maintenance of human ES and iPS cell pluripotency, as these cells are primed, unlike mouse ES, and require Activin A for their maintenance [132,133,134].

Generation of pluripotent stem cells may also be affected by perturbation of BMP activity. For instance, fibroblasts from fibrodysplasia ossificans progressiva patients carrying a mutation in the *ACVR1* gene, which resulted in hyperactivation of BMP-SMAD signalling, were found to have increased iPSC reprogramming efficiency [135]. The addition of exogenous BMP4 to cultures during the early stages of reprogramming was found to have a similar effect. Recently, modulation of the stiffness of the hydrogels on which fibroblasts were cultured during reprogramming to iPSCs was found to upregulate BMP2 and several genes involved in BMP signalling, as well as improve reprogramming of the cells. Increased hydrogel stiffness upregulated *Phactr3*, which then resulted in increased BMP2 and improved reprogramming efficiency [136]. How *Phactr3* causes an increase in BMP2 is not known, although *Phactr3* is known to associate with nuclear nonchromatin structure [137], where it might influence expression of genes involved in reprogramming. Alternatively, *Phactr3* may exert its effects by inhibiting polymerisation of actin within the cell, resulting in increased cell spreading [138]. Recent studies have shown that changes in cell shape can affect the distribution of BMP receptors on the cell membrane [139].

## 5. Uses of Pluripotent Stem Cells to Investigate the Role of BMP in Development

### 5.1. Advantages and Limitations of Human PSCs

Stem cells, particularly human-derived stem cells, are invaluable tools for studying development and disease mechanisms without the need for fetal samples, which are difficult to acquire. They provide the opportunity to study aspects of development that are specific to humans. Indeed, previous studies have demonstrated differences in development and disease mechanisms between humans and animal models [140,141]. For instance, in the case of the inner ear, development and maturation are nearly complete by approximately 36 weeks of gestation in humans [142,143], whereas in mice, the cells of the cochlea of the inner ear continue to develop and mature after birth until about the third post-natal week [144,145]. This highlights the need for human-specific models to study development. Moreover, patient-derived stem cells can be used to study development in a patient-specific or disease context, without the need for generating mutant cell lines that might not behave in the same manner or may fail to recapitulate some aspects of the disease. Gene correction of such patient-derived stem cells can also be used to correct mutations to investigate whether proper functioning of the gene is regained, opening the way for gene therapy treatments. For instance, patient-derived hiPSCs have recently been used to model mutations in *TMC1*, which are associated with a type of progressive hearing loss termed DFNA36 [146]. While differentiation of pluripotent stem cells to sensory hair cells was not affected by the TMC1 mutation, the morphology and electrophysiological properties of the derived hair cells were altered. Additionally, using CRISPR/Cas9 genome editing technique to generate an isogenic cell line, in which the mutated gene was corrected, resulted in the recovery of hair cell morphology and electrophysiology. Similar works have been done using patient-derived lines carrying mutations for several genes associated with hearing loss, including *USH2A* [147,148], *TRMU* [149], *ELMOD3* [150], *MYO7A* [151], and *AIFM1* [152], highlighting the potential strength of this approach for therapeutic genome editing.

In spite of recent developments and advances in stem cell technologies, numerous barriers must still be overcome before stem cells can reach their optimal potential in research and clinical applications. Many stem cell differentiation protocols result in batch-to-batch variability and also variation between labs, necessitating further refinements to produce more uniform and homogenous populations of the desired cell and tissue types being investigated. Furthermore, unlike studying development in animal models, such in vitro differentiation often occurs in isolation from other cell types, which might provide trophic and supportive factors beneficial to generating the cells under investigation. For instance, differentiation and development of inner ear hair cells require the support of the surrounding connective tissue and mesenchymal cells [153,154,155]. Indeed, otic mesenchyme cells comprise a diverse array of cell types that make up several important cell types in the inner ear, including spiral limbus fibrocytes and modiolar osteoblasts [156]. Since hair cells rely on neurons to transmit auditory signals to the brain, co-culturing stem cell-derived hair cells with spiral ganglion neurons should be considered to establish functional circuits. Additionally, the generation of vascularised organoids would be beneficial to enable the growth of larger and healthier organoids. Incorporation of such tissues in the form of co-cultures and assembloids can lead to the development of more robust and mature models, which have already shown promising results [157,158]. However, this may also complicate the system, especially if looking for populations of pure and mature cells with the intention of being able to transplant the generated cells into patients. Finally, different biomaterials should also be tested to investigate their roles and potential benefits in constructing more physiologically relevant 3D culture systems that better recapitulate the tissue microenvironment (reviewed in [159]), as remodelling of the extracellular matrix plays an important role during maturation of the cochlea (reviewed in [160]).

### 5.2. Otic Neurosensory Specification as a Model to Study BMP4 Signalling

Because BMPs are involved in many steps of inner ear development and are required at specific concentrations over precise durations [161], otic lineages provide an interesting model system to investigate the effects of BMP signalling during development. Human PSCs can be differentiated under either 2D or 3D culture systems to give rise to otic progenitors that express several of the markers and components of activation pathways found during early otic development, and eventually hair cells, supporting cells, and neurons in inner ear otic organoids that have been allowed to mature in long-term culture [17,18,20,22,24,162]. Recent advances in 3D-otic organoids have additionally been able to generate both cochlear and vestibular type hair cells [20], demonstrating the ability to finely control the generation of inner ear hair cells in such 3D-cell culture systems.

Mutations in some genes involved in BMP signalling are associated with hearing loss (Table 1). Nager syndrome is associated with hearing loss as a result of mutations in the *SF3B4* gene, which codes for a spliceosome that affects expression of Noggin and BMPs and may be directly involved in neural crest and otic development [163]. Mutations in chondroitin synthase 1 (*CHSY1*), involved in the synthesis of chondroitin sulfate, are characterised by limb malformations, short stature, and hearing loss [164], and studies in the inner ears of zebrafish larvae have found that *Chsy1* expression is similar to that of the BMP inhibitor *dan* and complementary to *Bmp2b* expression, suggesting a role for this gene in BMP signalling and otic development [165]. In cultures of mouse chondrocytes, knockdown of *Chsy1* resulted in increased BMP signalling, while overexpression of *Chsy1* reduced BMP signalling [166]. Whether similar effects of *Sf3b4*, *Chsy1*, and other genes potentially involved in BMP signalling (Table 1) can be observed in cultures of PSC-derived otic progenitors remains to be investigated.

One of the consequences of suboptimal BMP4 signalling during the early specification of otic progenitors under these pluripotent cell culture systems is the generation of off-target cell types, such as neurons and surface epidermis [8,20,24,171]. Current methods for detecting subtle differences in off-target differentiation are mostly restricted to immunolabelling and qPCR analyses for off-target genes, most of which are transcription factors. It has also been reported that the epithelial thickness of organoids after just 3 days in vitro can be used as a proxy to optimise BMP4 concentration in such cultures [24]. However, the link between BMP4 concentration and epidermal thickness is not clear, and this method requires the production and screening of many otic organoids. New methods that can allow for the rapid detection of off-target differentiation using fewer samples would enable researchers to detect such off-target effects more efficiently and gain a better understanding of the variations between different lineages, beyond the expression of transcription factors, for example, in the biochemical and metabolic properties of such in vitro differentiated cells.

Pluripotent stem cells also offer the opportunity to study the effects of BMP signalling at later stages of development. Several studies have reported conductive hearing loss in patients with mutations in the *NOG* gene, which encodes for the BMP antagonist Noggin, resulting from auditory-ossicle fusion [188,189,190,191,192,193,194,195,196,197]. As these patients exhibit additional symptoms, including bone and joint disorders and digital anomalies, patient-derived stem cells may facilitate the study of these mutations specifically in inner ear development. This approach could help determine at which stages in development these symptoms begin to appear, as well as follow disease progression and test the effects of potential therapeutics. PSC-derived otic organoids could also be used to investigate the role of different BMPs in cochlear and vestibular development in humans, as these organs have been shown to require differential BMP signalling in chick embryos [114].

## 6. Conclusions and Future Perspectives

Advances in stem cell research have greatly expanded our knowledge and understanding of development and the signalling pathways involved in developmental processes, while also prompting new questions and lines of investigation. Nevertheless, as the role of BMP signalling in the development of the inner ear and other tissues has demonstrated, further work is needed to better understand general human-specific developmental and disease pathways and mechanisms, rather than potentially batch or cell line-specific features. As signalling pathways other than BMP are likely to differ among cell types and perhaps among culture conditions, the starting state of stem cell cultures should be determined before initiating any cell differentiation protocol, in order to ensure that the optimal conditions for differentiation of the desired tissues are being met. New technologies could help to simplify the determination of endogenous levels of signalling molecules and signalling activity in cell lines, allowing for more robust and homogeneous cultures that better recapitulate in vivo conditions.

## Figures and Tables

**Figure 1 biomedicines-12-02262-f001:**
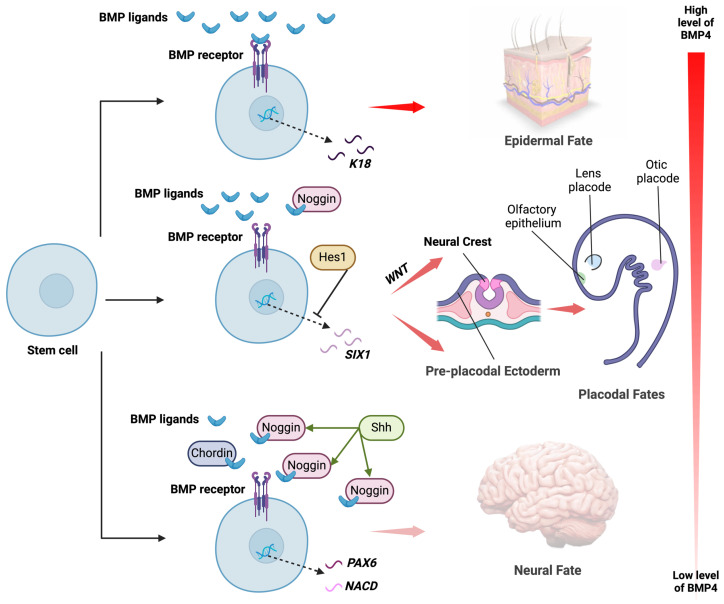
Effects of BMP concentration on fate of ectodermal cells to induce epidermal, placodal, neural crest, and neural derivatives. Exposure of pluripotent stem cells to different concentrations of BMP4 results in differentiation towards different cell fates via activation of various downstream genes. High concentration of BMP4 results in activation of genes such as *K18*, which causes cells to differentiate towards epidermal fate. Medium concentration of BMP4, which can be due to the presence of some inhibitors such as Noggin, causes activation of *SIX1* for differentiation towards pre-placodal ectoderm and subsequent placodal lineages including lens, olfactory, and otic placodes. However, in the presence of WNT, neural crest fate is induced. Activation of genes such as *Hes1* can have an inhibitory effect on this pathway. Low concentration of BMP4, which can be the result of high levels of Noggin due to Shh signalling or the presence of Chordin, results in activation of *PAX6*, *NCAD*, and other genes that result in neural fate. (Generated using Biorender.com, accessed 6 September 2024).

**Figure 2 biomedicines-12-02262-f002:**
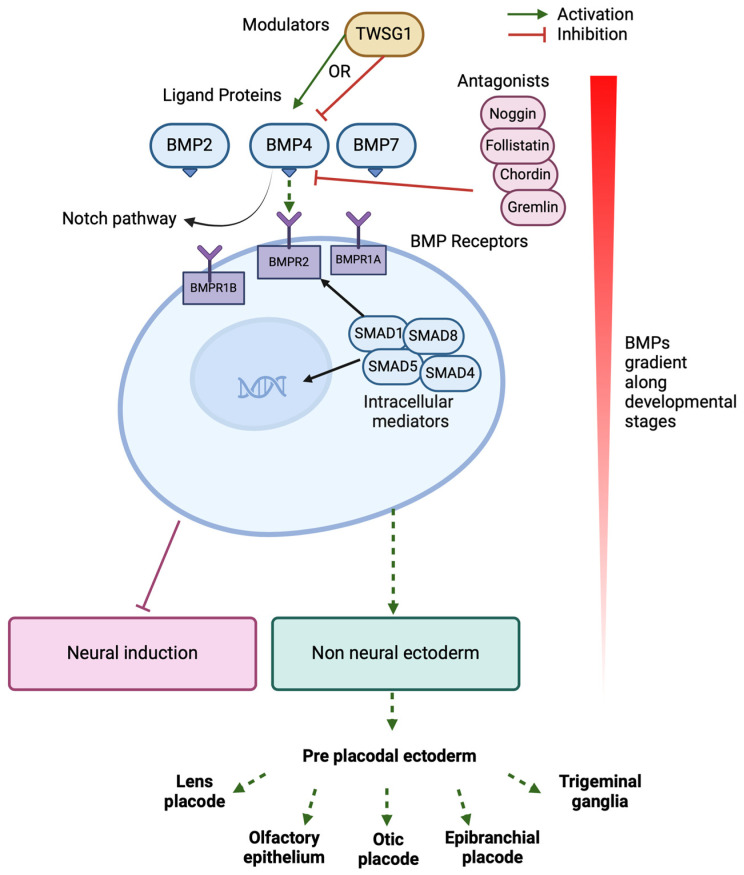
Overview of BMP signalling pathway and modulators during development of pre-placodal ectoderm. BMPs such as BMP4 bind to their receptors BMPR1A, BMPR1B, and BMPR2 on the cell surface, resulting in activation of SMADs which translocate to the nucleus to influence transcription of genes directing cell fate towards non-neural ectoderm/pre-placodal and subsequent placodal fates, and inhibiting differentiation towards neural fate. The presence of antagonists and modulators such as Noggin, Follistatin, Chordin, Gremlin, and TWSG1 alter the level of BMP activity on the cell and hence can also influence cell fate. (Generated using Biorender.com, accessed 6 September 2024).

**Table 1 biomedicines-12-02262-t001:** Genes involved in BMP signalling that are associated with hearing loss in humans.

Gene	Role in BMP Signalling	Inner Ear Deficits	Additional Symptoms	References
*ACVR1* (Activin A receptor type 1)	Type 1 BMP receptor	Sensorineural hearing loss; conductive hearing loss	Bone and skeletal disorders	[167,168,169,170]
*BMP2*	BMP ligand	Conductive hearing loss (otosclerosis)	Craniofacial, cardiac, and skeletal anomalies	[171,172,173,174,175]
*BMP4*	BMP ligand	Sensorineural hearing loss; conductive hearing loss (otosclerosis)	Eye, joint, and craniofacial disorders, renal dysplasia	[171,172,174,176]
*BMP7*	BMP ligand	Sensorineural hearing loss	Eye anomalies, developmental delay, scoliosis, cleft palate	[177]
*CHD7*	Promotes Col2a1 expression; regulation of BMPR1B expression	Sensorineural hearing loss; some conductive hearing loss due to enlargement of vestibular aqueduct	Vestibular dysfunctions, hypogonadotropic hypogonadism	[178,179,180]
*CHSY1* (Chondroitin synthase 1)	BMP inhibition	Sensorineural hearing loss	Facial dysmorphism, dental anomalies, digital anomalies, delayed motor development, delayed mental development, growth retardation	[164]
*COL2A1*	Binds BMPs	Sensorineural hearing loss	Short stature, bone and joint dysplasias, ocular problems	[181,182,183]
*GDF6* (Growth and differentiation factor 6)	Forms heterodimers with BMPs	Conductive hearing loss (otosclerosis); cochlear aplasia	Wrist and ankle deformities, tarsal–carpal fusion, vertebral fusion, speech impairment	[184,185,186,187]
*NOG* (Noggin)	BMP antagonist	Conductive hearing loss (stapes ankylosis and incus short process fixation)	Bone and joint disorders, digital and eye anomalies	[188,189,190,191,192,193,194,195,196,197]
*SF3B4* (Splicing factor 3B subunit 4)	Spliceosome that affects Noggin and BMP expression	Conductive, sensorineural, and mixed hearing loss	Craniofacial defects, limb defects	[163]
*SMAD4*	Downstream effector of BMP signalling	Conductive, sensorineural, and mixed hearing loss	Short stature, facial dysmorphism, muscular hypertrophy, cognitive delay	[198,199,200,201,202,203,204]
*TMEM53* (Transmembrane protein 53)	Inhibits BMP/SMAD signalling	Sensorineural hearing loss	Bone and eye disorders	[205,206]

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
