# Peer review of "Timing and Graded BMP Signalling Determines Fate of Neural Crest and Ectodermal Placode Derivatives from Pluripotent Stem Cells"

_biomedicines, 2024, doi:10.3390/biomedicines12102262_

Round 1

Reviewer 1 Report

Comments and Suggestions for Authors

The authors have a provided a comprehensive review of BMP signaling in in vitro cultures of neural crest and ectodermal placodal cell population from PSCs. The authors have covered significant literature describing/ citing the role of BMP signaling during differentiation of PSCs to various ectodermal cell types mainly focusing on recent advances in the field of PSC differentiation, organoid culture, and clinical relevance. The review covers wide areas of PSC differentiation into various cell types; however, it lacks in describing an in-depth role of BMP signaling in gene regulation during these processes. Review is also missing some citations of original research work on the role of BMP signaling during embryonic development relating to neural crest and placodal developmental. Authors have still covered many areas of PSC differentiation and role of BMP signaling during these processes. 

Comments on the Quality of English Language

None

Author Response

Comments 1: The review covers wide areas of PSC differentiation into various cell types; however, it lacks in describing an in-depth role of BMP signaling in gene regulation during these processes. 

Response 1: While there is some in-depth role of BMP signalling in gene regulation is the literature, information remains relatively scarce, and most of the currently available information is quite generic rather than for specific cell/lineage types. This is not overly surprising given that BMPs comprise a family of heterogeneous proteins that do not act in isolation and act on different receptors. Nevertheless, we have tried to address this issue by adding additional information where it could be found, such as induction of MAF by BMP signalling during lens development (line 177). 

Comments 2:  Review is also missing some citations of original research work on the role of BMP signaling during embryonic development relating to neural crest and placodal developmental.

Response 2: Additional citations have been added for original research on the role of BMP in neural crest and placodal developmental (e.g. line 120). We also created a new section for neural crest (section 2.4, with subsequent sections re-numbered accordingly) and added additional references. We feel that this has improved the manuscript and we hope that the reviewer agrees. 
There is a lot of original research on the role of BMP signaling during embryonic development; indeed the difficulty is trying to decide what to include! There are also a lot of detailed reviews that describe everything in a clear and concise manner, however, and we felt it would be more prudent to reference these where appropriate, rather than re-writing what has already been done. We also wanted the focus of the review to be on more recent developments, particularly with regards to the role of BMP signalling in stem cells and the use of stem cells to further study BMP signalling, rather than summarising the role of BMP in development. Of course, it is not possible to discuss the role of BMP signalling in stem cells without also discussing the role of BMP signaling during embryonic development, and it is difficult to get the balance right. We hope that we have at least been somewhat successful.

Reviewer 2 Report

Comments and Suggestions for Authors

The authors made a quite an effort to write the narrative review that deals with significance of BMP signaling in human pluripotent stem cells differentiation during development. The authors have done a great job doing research for this paper, which will facilitate work for scientists who want further investigate the deeper understanding of developmental pathways.

Here is a suggestion for improvement of the article:

In Introduction it is missed the basic information about Bone Morphogenetic Proteins (BMPs). 

Also, since it is a group of signaling proteins, it would be more suitable to use plural when mentioning BMPs proteins as a whole.

Author Response

In Introduction it is missed the basic information about Bone Morphogenetic Proteins (BMPs). 

We thank Reviewer 2 for the helpful feedback, which we feel has greatly improved the manuscript. The final paragraph of the Introduction, where further information about BMPs is given, is moved to line 50 of the original submitted manuscript to become the third paragraph of the Introduction, and the text has been adapted accordingly to help it read better. We did not want to go into too much detail about BMPs in the introduction, as they are discussed in more detail later on in the manuscript, but we hope that by introducing them earlier in the Introduction that the reader has a better understanding of them before reading further. We also think that this has improved the flow of the Introduction section. 

Also, since it is a group of signaling proteins, it would be more suitable to use plural when mentioning BMPs proteins as a whole.

The reviewer is also correct that BMPs comprise a group of proteins, so we have changed "BMP" to "BMPs" in several instances where appropriate (e.g. original lines 19, 78, 133, 139, 149, 177, 203, 287, 321, 599), and adapted the text accordingly.